# Mental Health of People with Intellectual Disabilities Living in Residential Care before, during, and after Lockdown

**DOI:** 10.3390/bs13080695

**Published:** 2023-08-21

**Authors:** María Dolores Gil-Llario, Irene Díaz-Rodríguez, Olga Fernández-García, Verónica Estruch-García, Mar Bisquert-Bover, Rafael Ballester-Arnal

**Affiliations:** 1Department of Developmental and Educational Psychology, Faculty of Psychology, University of Valencia, 46010 Valencia, Spain; dolores.gil@uv.es (M.D.G.-L.); idiazro@alumni.uv.es (I.D.-R.); olga.fernandez-garcia@uv.es (O.F.-G.); marbisbo@alumni.uv.es (M.B.-B.); 2Department of Basic and Clinical Psychology and Psychobiology, Faculty of Health Sciences, Jaume I University, 12006 Castellón, Spain; rballest@uji.es

**Keywords:** COVID-19, lockdown, mental health, people with intellectual disabilities, quality of life

## Abstract

Background: The impact of the COVID-19 on the well-being of people with intellectual disabilities (PID) has been little studied. Methods: We analyzed its impact with a cohort study quantitatively analyzing anxiety, depression, organic symptoms, quality of life, and support needs in 24 PID, aged 19–74 years ($\bar{x}$ 40, σ = 13.09), living in a residential center, before, during, and after the pandemic. Results: Their mental health improved unexpectedly at the onset of the lockdown although there was an increase in organic symptoms. But, with the progress of the lockdown, their mental health deteriorated drastically. On the contrary, as expected, their quality of life and support needs worsened from the beginning of the lockdown until the country returned to normality, a time when there was a general recovery, without reaching pre-pandemic levels. These results show that the mental health of PID was affected differently to that of people without intellectual disabilities.

## 1. Introduction

On 11 March 2020, the rapid increase in coronavirus cases caused by the SARS-CoV-2 virus prompted the World Health Organization to declare a global public health emergency [1,2]. Undoubtedly, the situation created by the pandemic had a significant impact on the health of the world’s population [3]. The direct effects of the virus included symptoms such as fever, cough, fatigue, and difficulty breathing [4], with severe cases leading to complications such as pneumonia, acute respiratory distress syndrome, organ failure, and even death [5].

However, the severity of the disease and the consequences of the state of emergency do not affect everyone equally, with individuals with intellectual and developmental disabilities (IDDs) being one of the most vulnerable population groups in this situation [6]. Even before the pandemic, people with intellectual disabilities (PID) already required additional healthcare support due to their high rates of comorbidity and increased health risks [7]. Thus, the negative consequences of the coronavirus have been more pronounced for this population [8], as they are more vulnerable to the effects of the virus [9,10]. Studies have shown that PID have a higher risk of hospitalization and mortality due to COVID-19 infection compared to those without intellectual disabilities [11].

Furthermore, PID have cognitive limitations and many of them need support to carry out activities of daily living [12]. For this reason, PID would likely need to rely on others to protect them safe from infection during a pandemic, as their cognitive abilities may make it difficult for them to adhere to public health measures to reduce the spread [13]. Additionally, caregivers may find it challenging to support people with hygiene activities while maintaining safety measures, considering the sick leave that occurred at that time [7].

However, the measures implemented affected PID in multiple ways. Muñoz-Bravo and de Araoz-Sánchez-Dopico [14] pointed out that in situations where healthcare resources are scarce, as was the case during the pandemic, society, including healthcare professionals, tends to undervalue the quality of life of PID, which can lead to discriminatory decisions towards this group. This seems to be supported by the findings of Doody and Keenan [15], who noted in their review that, due to the measures implemented to control the spread of the virus, access to education and intervention services, such as mental health care services, was restricted for them. Moreover, PID follow specific routines and need to be prepared for any changes [16]. However, the need to respond quickly to the pandemic situation meant that their academic and recreational activities had to cease [17]. Moreover, PID who were employed, which helps them feel self-sufficient, lost their jobs because of the pandemic [18,19]. This disruption led to feelings of anger and frustration [20].

Several studies conducted in the first months of the pandemic have revealed that fear, worry, and stress were prevalent among individuals with IDDs as a result of the situation described [21]. The ongoing pandemic has created a sense of uncertainty about the future, further intensifying feelings of anxiety and stress among Spanish individuals with IDDs and their families [22].

Research has also highlighted the differences between PID living in residential centers and those living with their families. Thus, PID living in residential centers are more at risk of mental disorders than those living with their families in general conditions [23,24]. These differences between PID living in residential centers and those living with families were more evident during institutionalization [25]. Thus, studies carried out in Spain indicated that residential users experienced a greater lack of contact with their family and friends, mainly due to the reported suspension of visits to residential users and new admissions [26], which together with difficulties in accessing technology [6], contributed to increased levels of stress and emotional deterioration among residential users [17,27], as well as a greater sense of isolation due to not being able to see their friends and partners for longer periods of time [28]. Similar results were found by Amor et al. [29], who described that 60% of the sample reported higher levels of fear and anxiety during lockdown, particularly those living in residential centers. Another aspect highlighted was the impact of the suspension of work activities, as it is common for residents of community homes to work outside the center, which helps them to feel self-sufficient and increases their participation in the community [27].

However, it should be noted that some positive outcomes have been observed in other research, such as greater perceived freedom to choose personal goals [30]. On a positive note, staff adherence to health protocols and the implementation of structured daily routines contributed to a reduction in medication errors [31].

These contradictory results may be due to the fact that many of the investigations were carried out without taking into account changes in the measures implemented during the pandemic. So, studies that analyze the impact on mental health and quality of life in each country according to how the pandemic has evolved over time are needed [21,32]. In Spain, at the beginning of the pandemic, severe restrictions were imposed on all populations. However, three months later, these measures began to be relaxed for people not living in residential centers, with some daily activities being resumed, although without reaching a state of normality. But the return to normality in residential centers was slower, given the high vulnerability of this population, and the lockdown of this group was prolonged for almost two years after the onset of the COVID-19 pandemic [28].

Given the significant impact of COVID-19 on PID and the prolonged restrictions experienced by those living in residential settings, it is imperative to assess the psychosocial impact by analyzing the different phases of the pandemic separately. The main objective of this study is to assess the negative impact of lockdown measures on the mental health, quality of life, and support needed by PID living in residential centers. Specifically, the study examines mental health indicators, including anxiety, depression, and organic symptomatology. In addition, it explores changes in quality of life and the level of support needed at different stages of the lockdown. Mental health and quality of life indicators are expected to decline during the pandemic, but to recover when the situation returns to normal. The level of support required during the pandemic is expected to be higher than before the pandemic levels and to gradually decrease as conditions improve.

## 2. Methods

### 2.1. Study Design

A cohort study with follow-up was designed to quantitatively assess and compare the level of anxiety, depression, organic symptoms, support needs, and quality of life of the participants in this study at the four points in time when this information was collected.

### 2.2. Setting

First, we contacted the psychologist of a residential center for PID in the community of Andalusia (in the south of Spain), who joined the project after obtaining the agreement of the center’s management.

The participants were evaluated 4 times (Table 1), temporarily spaced from each other, as shown in Figure 1: (1) in a situation of normality before the COVID-19 pandemic and the implementation of any restrictive measures (TIME 1: Pre-COVID-19 normality); (2) at the beginning of the lockdown, which led to the closure of the occupational center and restrictions on entering and leaving the residential center (people inside and outside the center). The residential users had been confined in a single bubble group in which all of them interacted with each other with a certain degree of freedom. In addition, given the abruptness of the decision, it was decided to make the timetable and activities more flexible (TIME 2: Lockdown in residence with the flexibility to interact between them); (3) during the lockdown, a period in which the restrictions within the center were more extreme than in the past. In addition to the previous measures, the users of the residence had been divided into stable groups, which is to say that each one could only interact with people living on the same floor. Specifically, the men lived on one floor, the older women on another, and the younger women on yet another (TIME 3); and (4) at the stage when the restrictions were eased. A sort of normality was restored, restarting the activity of the occupational center, allowing them to interact with other users of the occupational center, who lived alone or with families, although the division of residence users into small groups was maintained (TIME 4).

Regarding data collection, the first measure (TIME 1) was already included in the participants’ files, and the rest were obtained by following the usual registration procedure according to the mental health levels of the residential users. The second and third measures (TIMES 2 and 3) were collected online due to restrictions prohibiting entry into the residence. The last moment of the assessment (TIME 4) was carried out in person, but always during sessions with the psychologist. There was no loss of samples, given the lockdown restrictions. This situation reduced the possibility of contaminating variables appearing. Moreover, in the use of the different measuring instruments, the guidelines and instructions contained in the respective manuals were followed, respecting the established time intervals for re-administration, which ensure the validity and reliability of the scales used.

Throughout the process, all the rules set out in the Helsinki Declaration regarding ethical procedures and data protection were followed, while anonymity was fully guaranteed. All subjects gave informed consent for inclusion before participating in the study. The protocol was approved by the Ethics Committee of the University of Valencia.

### 2.3. Participants

The sample consists of 24 PID (45.8% men and 54.2% women). The age of the participants ranged from 19 to 74 years old ($\bar{x}$ = 40 years, σ = 13.09). Some 70% of residence users had moderate intellectual disability, compared to 29.2% who had a mild one. Table 2 shows the main descriptive dates.

To determine eligibility for the research, we established the following inclusion criteria: (a) have a diagnosis of mild or moderate intellectual disability; (b) be over 18 years of age; (c) be in a residential regimen; (d) have been assessed by the center on the variables of interest prior to the pandemic.

### 2.4. Measurement

The INICO-FEAPS Scale of Comprehensive Assessment of the Quality of Life of People with Intellectual or Developmental Disabilities [33] consists of 72 items with a four-option Likert scale response format: never, sometimes, frequently, and always. The dimensions of quality of life evaluated were: emotional well-being, interpersonal relationships, material well-being, personal development, physical well-being, self-determination, social inclusion, and rights. The reliability analysis found α ranging between 0.65 and 0.89 in the different dimensions of INICO-FEAPS Scale in this study.

San Martín Scale of Assessment of the Quality of Life of People with Significant Disabilities [34] provides information on the most important areas of the life of a PID from the perspective of an external observer who knows them well. The instrument consists of 95 items organized around the eight dimensions of quality of life which are the same as those of the INICO-FEAPS. The items are answered using a four-point Likert response format: never, sometimes, frequently, and always. The internal consistency for all dimensions of San Martín Scale ranged between r = 0.68 and r = 0.83 in this study.

Both scales, the INICO-FEAPS Scale and San Martín Scale, are equivalent to each other [34]. The INICO-FEAPS Scale is an assessment tool based on the same theoretical model and dimensions (factors) as the San Martín Scale. The San Martín Scale was developed due to a lack of assessment tools for quality of life in Spain for people with higher disabilities.. So, the INICO-FEAPS Scale was used to assess the 7 participants with mild PID given their ability to respond to self-reports, and the San Martín Scale was used to assess the 17 individuals with moderate PID. According to the theoretical model on which these two scales are based, which assumes that individuals live in complex social systems made up of and influenced by different contexts, both scales are useful instruments to identify change in the readministration time period used in the study [35].

Supports Intensity Scale (SIS), adapted to Spanish [36,37] is a functional assessment of the PID’s needs, focusing on the type of support they require rather than the deficits they have. This is applied through the semi-structured interview format. Through the subscales, 10 supports are measured in 57 life activities, relating to the areas of their home living; community living; life-long learning; employment; health and safety; social activities. This scale has been applied to the whole sample, and the validity of its reapplication has been ensured by meeting the indicators that the original authors felt should guide further administrations of the scale (e.g., changes in health status and/or the occurrence of problem behaviors) [38]. The reliability of the different subscales was higher than r = 0.90 (home living, r = 0.98; community living, r = 0.98; life-long learning, r = 0.98; employment, r = 0.97; health and safety, r = 0.97; social activities, r = 0.97), and for the total scale score (r = 0.92) in this study.

Diagnostic Assessment for Severely Disabled DASH-II [39], adapted to Spanish [40], consists of 84 items measuring neurological and other mental disorders. The instrument is divided into 13 subscales, although our study focused on anxiety, mood disorder (depression), and organic syndromes. For this study, the frequency of occurrence of the main symptoms of these disorders was taken into account, ensuring that this measure identified change at each time period of readministration [41]. This scale has been completed by all participants. The internal consistency of the DASH-II dimensions, measured as Cronbach’s alphas (r), was r = 0.92 (anxiety), r = 0.89 (depression) and r = 0.94 (organic syndromes).

### 2.5. Bias

To ensure the feasibility of this research and for practical and accessibility reasons, convenience sampling was used to select the location where to carry out the study. The selection of a specific occupational center in Andalusia was convenient because it was easily accessible and allowed for a more efficient data collection process. In addition, despite the small sample size, simple random sampling was used to reduce possible bias in the selection of participants.

### 2.6. Study Size

The total sample in the present study, selected by simple random sampling, consisted of 24 participants who attended one occupational center for PID. Caregivers extracted an alphabetical list of all residence users who met the criteria. Then, one of the main researchers of this project used an online application that randomly extracts numbers, repeating the process up to 24 times.

### 2.7. Statistical Methods

First, descriptive statistics were obtained to characterize the sample. Means ($\bar{x}$) and standard deviations (σ) were extracted for numerical variables and percentages (%) for categorical variables.

Secondly, to analyze whether changes in mental health, quality of life, and support required by PID occur during the lockdown, means ($\bar{x}$) and standard deviations (σ) were extracted at four assessment times. Also, to determine whether the differences between the mean scores obtained at each of the assessment times were statistically significant, a four-way ANOVA analysis of variances was performed with repeated measures on one factor, asking for the value of the partial eta squared (η^2^). The sphericity condition had previously been tested, using Mauchly’s sphericity test.

All analyses were performed using frequency scores for DASH and percentile scores for INICO, San Martín, and SIS scales, and using the SPSS 26 statistical package.

## 3. Results

For the different mental health subscales analyzed in the DASH-II (anxiety, depression, and organic syndromes), mean scores appear to follow a similar trend across the different assessment times, as shown in Table 3.

As for depressive and anxiety symptoms, mean scores decreased at TIME 2 of the lockdown, during which residential users were able to interact freely and flexibly with each other (anxiety: 1.79 ± 1.91; depression: 4.21 ± 4.21) compared to TIME 1 of the pre-COVID-19 normality (anxiety: 2.0 ± 2.33; depression: 4.46 ± 4.42). However, their mean scores increased at TIME 3, when residential users were divided into stable groups (anxiety: 2.67 ± 3.41; depression: 5.12 ± 5.17). These scores then decreased again at TIME 4 during the period of quasi-normality (anxiety: 2.12 ± 2.27; depression: 4.5 ± 4.59), although they did not return to the pre-COVID-19 state of normality. Despite this, these differences between scores at different times of the assessment are not statistically significant in the repeated measures analysis for either anxiety (F = 1.24, *p* = 0.276, η^2^ = 0.05) or depression (F = 0.8, *p* = 0.38, η^2^ = 0.03).

For symptoms related to organic syndromes (such as increased restlessness, difficulties in focusing attention or remembering things that the individual used to know, slower response times, or rapid changes in mood), the evolution of mean scores is different. At TIME 2, when residential users were under lockdown with the flexibility to interact with each other, their scores continued to rise. This upward trend persisted at TIME 3 when residential users were divided into stable groups (T1: 2.42 ± 1.86; T2: 3.26 ± 3.24; T3: 3.79 ± 3.5). However, the mean score at the last moment of assessment when they were in the quasi-normality situation (T4: 2.83 ± 2.53) is lower compared to TIMES 2 and 3. These differences in scores at different assessment times have been statistically significant in the repeated measures analysis (F = 5.26, *p* = 0.032, η^2^= 0.13).

In terms of quality-of-life indices, there was an upward trend in self-determination scores at the beginning of the lockdown with the flexibility to social interaction (T2: 47.25 ± 26.25). However, these scores seemed to stabilize when residential users were divided into stable groups (T3: 44.37 ± 27.16) and gradually increased with the restoration of quasi-normality (T4: 44.63 ± 26.71), although they did not reach pre-COVID-19 levels of normality (T1: 46.33 ± 24.82). Repeated measures analysis reports no statistically significant differences between the self-determination scores at the different assessment times (F = 0.03, *p* = 0.868, η^2^ = 0.001).

Emotional well-being declined significantly during the early phases of the pandemic, when residential users were under lockdown with the flexibility to interact with each other (T2: 41.58 ± 23.03). However, it increased during the division of residential users into stable groups (T3: 42.79 ± 24.04) and remained stable afterward (T4: 42.67 ± 24.56), but still did not reach the levels of normality (T1: 45.38 ± 23.67). Repeated measures analysis reports no statistically significant differences between the emotional well-being scores at the different assessment times (F = 0.29, *p* = 0.592, η^2^ = 0.01).

In contrast, residents’ sense of social inclusion initially increased when they were all together during the lockdown (T2: 44.37 ± 28.31), compared to pre-COVID-19 normality (T1: 42.63 ± 24.91). However, the feeling of social inclusion declined with the division of residential users into stable groups (T3: 40 ± 31.15) and continued to decline during the period of quasi-normality (T4: 39.92 ± 27.39). A similar trend was observed in interpersonal relationships, where scores increased during the lockdown with flexibility to interact (T2: 43.13 ± 28.98), slightly during the advanced stage of lockdown (T3: 43.71 ± 33.12) and in the situation of quasi-normality (T4: 44.67 ± 26.8), compared to the pre-COVID-19 normality (T1: 41.29 ± 28.81). For both variables, repeated measures analyses report no statistically significant differences between scores at the different assessment times (social inclusion, F = 0.17, *p* = 0.682, η^2^ = 0.007; interpersonal relationships, F = 0.59, *p* = 0.449, η^2^ = 0.02).

Analyzing the support needs of the participants at different assessment times (Table 4), it is observed that, in general, the support required increased from the time of the lockdown with flexibility to interact (T2: 78 ± 23.1; T3: 82.04 ± 19.19), and only decreased when the situation returned to normality or quasi-normality (T4: 77.71 ± 23.81). Repeated measures analysis revealed statistically significant differences between the scores at the different assessment times (F = 4.69, *p* = 0.041, η^2^ = 0.18).

The mean scores for all the variables analyzed, including support needs at home, health, safety, and social aspects, follow a similar trend to the one mentioned above. The support required increased from the beginning of residential lockdown with flexibility for social interaction (home living, T2: 79.92 ± 19.64; T3: 81.04 ± 19.21; health and safety, T2: 78 ± 24.87; T3: 79.21 ± 23.79; social activities, T2: 75.38 ± 25.73; T3: 77.92 ± 22.9), and with the recovery of greater normality (quasi-normality), the support required decreased. For the variable “Home Living”, repeated measures analyses report statistically significant differences between scores at the different assessment times (F = 5.59, *p* = 0.027, η^2^ = 0.19), while for the variables “Health and Safety” and “Social activities”, these differences were not statistically significant (health and safety, F = 2.48, *p* = 0.129, η^2^ = 0.09; social activities, F = 1.58, *p* = 0.221, η^2^ = 0.06).

In terms of the support needed to engage in activities outside the home, assessed through the community life and employment subscales, the support requirements increased during the lockdown in the residential center with the flexibility to interact (community living, T2: 74.08 ± 27.52; T3: 73.71 ± 27.89; employment, T2: 77.83 ± 24; T3: 76.83 ± 25.59). However, as measures were relaxed and conditions moved towards greater normality (quasi-normality), the support needed to carry out activities in these contexts decreased (community living, T4: 73.25 ± 29.1; employment, T4: 73.58 ± 28.61). In this sense, repeated measures analyses report statistically significant differences between the scores obtained at the different assessment times for the variable “Employment” (F = 5.47, *p* = 0.028, η^2^ = 0.19), but not for “Community living” (F = 0.78, *p* = 0.387, η^2^ = 0.03).

## 4. Discussion

Through this study, our aim was to present to the scientific community the potential consequences of lockdown for PID, especially on their mental health, quality of life, and support needs.

Initially, studies of the general population during lockdown [42,43] led us to hypothesize that PID would experience more pronounced negative effects on their physical and mental health compared to the rest of the population during the initial phase of complete mobility restriction (TIME 2). Paradoxically, however, our study results suggest that the mental health of PID who were confined to their residential center (not within the family unit) improved at this time. This contrasts with other studies of PID, which reported high rates of depression, anxiety, psychological distress, and loneliness during the early stages of the COVID-19 pandemic [6].

These findings can be understood considering the limited access to information that PID had during the lockdown [27], which may have resulted in a reduced understanding of the magnitude of the problem [44]. This, in turn, may have contributed to a decrease in anxiety and uncertainty compared to those who were exposed to contradictory and alarming media information. Additionally, the opportunity for PID to freely interact with each other during this time increased their sense of social inclusion and social support, which is a mediator of individual stress and life satisfaction [18]. Moreover, more flexible routines and reduced obligations further improved their mental well-being.

However, a significant decline in mental health was observed when residential users were divided into smaller, independent stable groups without flexibility, between June and September, limiting their social interaction and social support [30]. These results are consistent with similar findings in the general population, where the psychological consequences of lockdown and the ongoing global pandemic have been reported [45,46]. In vulnerable groups such as PID, the lockdown measures can induce or exacerbate feelings of loneliness, which in this case may have contributed to higher levels of depression and anxiety, ultimately affecting their overall quality of life, as demonstrated in other studies [21,29,47]. These findings highlight the importance of considering the specific needs of PID in lockdown situations and providing appropriate support to mitigate the negative impact on their mental well-being and quality of life.

Furthermore, as expected, there was a gradual improvement in the mental health of PID as the measures were eased and a state of quasi-normality was reached. However, it is important to note that their mental health has not fully returned to pre-pandemic levels. This can be attributed to the prolongated duration of the restrictions, and the losses experienced, such as the loss of their jobs [19], relatives, and friends due to COVID-19 [28].

In terms of their support needs, the disruption of their daily routines and the implementation of unprecedented health and safety measures since the early stages of the pandemic have resulted in an increased demand for support. This increase in needs, coupled with the loss of support staff due to infections and quarantines [48], may explain the relatively low levels of quality of life indicated by their scores [13]. As indicated in a study conducted by Crespo [6], the situation brought about by COVID-19 has highlighted the need to increase the number of professionals to provide better support to PID.

Considering that this specific group faces a particular reality in which mental health problems are more prevalent [49], their vulnerability to the biopsychosocial impact of the disease and to the measures implemented to control it is heightened [50,51]. It is therefore crucial to monitor their progress closely and apply interventions according to the results in order to mitigate any potential sequelae that may hinder their overall development and daily functioning.

It is important to acknowledge some limitations of our study. Firstly, the study was conducted on small sample size, mainly due to the challenges of accessing this population, particularly during the pandemic [52]. In this sense, the retrospective power calculation does not reach 70% (1 − β = 0.67), which should be taken into account when generalizing the findings and could be the reason for the non-significance of some of the results. In addition, a control group was not included, which requires caution in the interpretation of our results. However, in contrast to other research, our study provides valuable additional data, including the evolution of participants through the different stages of the pandemic, given the multiple waves that were conducted. This provides a more complete understanding of the experiences of PID residing in residential centers throughout the pandemic situation.

## 5. Conclusions

In conclusion, the findings indicate that the mental health of PID was not negatively affected initially and even showed some improvement during the initial phase of the lockdown. Unlike the general population, PID experienced greater freedom and flexibility in their schedules, allowing them to be together. However, as their lockdown was prolonged and they were separated from each other, interacting only with members of their bubble group, their mental health worsened significantly. It is important to note that the mental health levels of the PID have not fully recovered to pre-lockdown levels, as this group continues to be subjected to strict measures while the general population returns to normality.

These results raise awareness of the possible consequences of overprotection on the mental health of PID. It is worth mentioning that even some individuals without intellectual disabilities have needed psychological support to cope with the mental challenges arising from the pandemic. Therefore, we strongly recommend that attention be paid to possible psychological sequelae that may have been intensified by the pandemic and the current post-pandemic conditions. The prolonged social restrictions imposed on PID have had a greater impact on this group compared to the population without intellectual disabilities.

## Figures and Tables

**Figure 1 behavsci-13-00695-f001:**
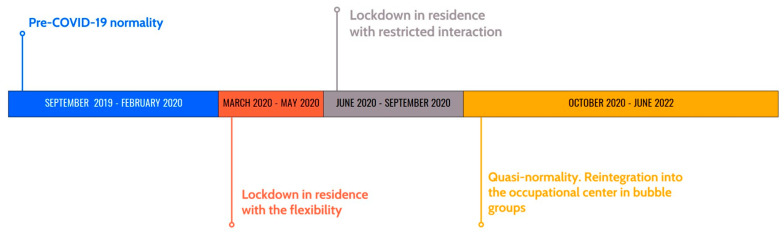
Timeline of assessment times.

**Table 1 behavsci-13-00695-t001:** Characteristics of assessment TIMES.

	Assessment	Description
TIME 1 19 September–20 February	T1: 20 February	Pre-COVID-19 normality
TIME 2 March–20 May	T2: 20 May	Lockdown in residence with flexible schedules, activities, and freedom of interaction between all the residential users.
TIME 3 June–20 September	T3: 20 September	Residents divided into stable groups. Lockdown in residence with limited interaction between people on the same floor (only men/only young women/only older women)
TIME 4 21 October–22 June	T4: 22 June	Quasi-normality. Reintegration into the occupational center in bubble groups. Interaction with family home users is allowed for two weeks in June, but the new restriction to the bubble group was reinstated in mid-June 2021. As a result, data collection was extended until the restrictions were reduced to a minimum, which took one year for this phase.

**Table 2 behavsci-13-00695-t002:** Sample characteristics.

Variables	$\bar{x\;}$± σ or % (*n*)
Sex	
Women	54.2% (13)
Men	45.8% (11)
Age	40.17 ± 13.09
Level of Intellectual Disability	
Mild	29.2% (7)
Moderate	70.8% (17)

**Table 3 behavsci-13-00695-t003:** Mean and standard deviation of the total score of anxiety, depression, and organic syndromes at different times of assessment.

	T1. ^1^		T2. ^2^		T3. ^3^		T4. ^4^	
	$\bar{x}$	σ	$\bar{x}$	σ	$\bar{x}$	σ	$\bar{x}$	σ
Anxiety	2.04	2.33	1.79	1.91	2.67	3.41	2.12	2.27
Depression	4.46	4.42	4.21	4.21	5.12	5.17	4.50	4.59
Syn. Organic	2.42	1.86	3.26	3.24	3.79	3.5	2.83	2.53

Note: ^1^ Pre-COVID-19 normality, ^2^ lockdown in residential center with the flexibility to interact between the residential users, ^3^ lockdown with the split of residential users into stable groups, ^4^ quasi-normality (bubble groups).

**Table 4 behavsci-13-00695-t004:** Mean and standard deviation of the total score in support needs and in the variables that make it up at the different times of assessment.

	T1. ^1^		T2. ^2^		T3. ^3^		T4. ^4^	
	$\bar{x}$	σ	$\bar{x}$	σ	$\bar{x}$	σ	$\bar{x}$	σ
Support Needs Total Score	77.83	22.78	78	23.1	82.04	19.19	77.71	23.81
Home Living	78.08	21.08	79.92	19.64	81.04	19.21	79.71	20.84
Community Living	73.42	28.3	74.08	27.52	73.71	27.89	73.25	29.10
Employment	73.5	27.13	77.83	24	76.83	25.59	73.58	28.61
Health and Safety	76.42	24.02	78	24.87	79.21	23.79	77.29	25.61
Social activities	75.08	22.94	75.38	25.73	77.92	22.9	77.17	23.82

Note: ^1^ Pre-COVID-19 normality, ^2^ lockdown in residential center with the flexibility to interact between the residential users, ^3^ lockdown with the split of residential users into stable groups, ^4^ quasi-normality (bubble groups).

## Data Availability

The data presented in this study are available on request from the corresponding author. The data are not publicly available due to privacy.

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
