# Peer review of "Mental Health of People with Intellectual Disabilities Living in Residential Care before, during, and after Lockdown"

_behavsci, 2023, doi:10.3390/bs13080695_

Round 1

Reviewer 1 Report (Previous Reviewer 2)

It is deemed that the fundamental issues were (and may not be possible) to be addressed under current study design. 

None.

Author Response

Point 1: It us deemed that the fundamental issues were (and may not be possible) to be adressed under current study design.

Response 1: First of all, we would like to thank you for the feedback you have given us on our paper. Thank you for the time and effort you put into your review.

We would like to reflect with you on the circumstances in which this research is being carried out, which, to some extent, testifies to the difficulty of carrying it out successfully and the importance of making its results known to the scientific community and to society. The lockdown, which began in March 2020, brought about a drastic change at all levels, forcing everyone to change their habits and the way they interact. Health centres, such as residential centres, prohibited the entry and exit of any person who was not essential for the basic functioning of the centre. This obliged the professionals not only to adapt their work in order to care for the users of the residence while maintaining a safe distance, but also to provide them with maximum support so that the deprivation of contact with the outside world would not have an irreversible and severe impact on their mental health. This made it even more difficult to carry out research in an already difficult to access context, such as residential centres for people with intellectual disabilities. In addition, the complications that are assumed in any research with people with intellectual disabilities (e.g. limited cognitive capacity, limited validated measurement instruments for this population and/or difficulty in obtaining permissions from primary caregivers) must be taken into account, making it a slower and more complicated process. In this context, a cohort study was carried out which has been evaluated over several years without loss of samples. It is true that this research has some limitations that have been exposed in the "Discussion" (lines 364-372) because we believe it is absolutely necessary for the future reader to be aware of them. However, the fact that the data have been collected in several waves, which allows us to show the evolution of a group of people with intellectual disabilities throughout a period of time of exceptional circumstances, such as the pandemic situation we were experiencing, we believe that it makes it a very rich work and of great importance for future readers looking for explanations about the state of mental health of this group at present or possible changes in the way they relate to each other.

We would also like to inform you of the extensive improvements we have made to our paper, thanks to your comments and those of the other reviewers. In this regard, we have expanded the information on the assessment process followed and the design features of this study ("Setting" section). We have also added the results of repeated measures analyses for all variables (“Results” section). And, of course, following your recommendation, we have carried out a thorough revision of the English language, with the advice of a native English translator.

Of course, we remain at your disposal for any concerns/questions you may have about our research and our paper.

Reviewer 2 Report (New Reviewer)

Dear Authors

I have enjoyed reading your article which adds to the growing literature on how people with intellectual disability survived COVID. Your aims are well laid out to assess the negative impact of lock down measures on mental health, quality of life and support for people living in residential care centres.  You selected four distinct stages of the lockdown process which will resonate with services  across the globe. 

Some points requiring clarification include

1. The demographic details although of interest are not used for any statistical analysis. Age and level of ID do give a general background however what is the purpose of indicating sexual preference when it  is not used in any form of analysis?

2. Different quality of life scales are used depending upon level of disability which does therefore make sense of including  level  in the sample characteristics.  However when it comes to use of scales there  are  no details on the numbers of participants  who were administered which scale. This needs to  be included in respective sections that describe the measures.

3.  For each scale some  detail on its reliability and validity and also what evidence is there that each measure  has identified change within the time period  of re-administration  used within the study?   Time periods between each moment of readministration are very short for the first three moments.     Therefore  in the absence of any qualitative data describing  behaviour change,  evidence that  such  measures  have shown change within 2 month periods in other studies is needing be uncovered or defended.  Reference to the respective manuals of each measure may be helpful also. 

4. In terms of statistical results only the equation for those assessment moments that show significant difference in the analysis of repeated measures are given. Details of where significant difference was not reached the full statistical  analysis details  also need to be given 

 At this stage I have recommended  major revision to give you time to respond  to the timing issue of Assessment MOMENTS  ( See Table 4) .    

Author Response

Reviewer 3 Report (New Reviewer)

Dear editor, thank you for the opportunity to review this manuscript. It is a compelling study which examined potential differences in mental health, QoL and needs.

My greatest concern is the sample size. It is unclear how many were initially recruited, and how many subsequently dropped out. Was there a sample size calculation, and was the study of enough power? If yes, kindly show the post-hoc analysis as justification. Thank you.

There is moderate need for language editing.

Round 2

Reviewer 2 Report (New Reviewer)

Thank you for your comments  in response to my feedback The article reads very well and will be well received by the field of mental health and intellectual disability.  Best wishes for future owri in the area of intellectual  disabilities 

Author Response

We are pleased that you find that our paper meets the high-quality standards of "Behavioral Sciences" and is ready for publication. Furthermore, it is a pleasure for us to read that our paper can make an important contribution to the field of mental health and disability, of course, our ultimate goal.

We thank you for taking the time to review our article, as well as for your comments, which have helped us to improve the manuscript.

Reviewer 3 Report (New Reviewer)

I would like to thank the authors for the extensive input on why the sample is small. Thank you. Still, I would like to see the post-hoc sample size calculation. This I believe is a responsible step, to inform readers that the non-significance of the results may be due to the sample size rather than other reasons. Without this, the paper would not be telling the "whole" story, and is therefore potentially misleading. I ask the authors to seriously consider presenting the analysis requested

Author Response

First of all, we would like to thank the reviewer for appreciating the effort we have made to improve our paper and to convey the reasons why we did not have a larger sample size.

We also fully agree with the reviewer that the inclusion of the retrospective power calculation of the sample size is necessary to avoid misleading future readers in the interpretation of the findings presented. In this sense, using G*Power software, we calculated the post-hoc sample size calculation at the α = 0.05 level, and with a mean effect size. The results of this calculation have shown that the statistical power with our sample size is moderate, which, as intuited by this reviewer, could be behind the lack of significance of the results obtained.

We have included this information in the limitations section of our study so that the reader has all the information, thus avoiding possible misinterpretation of the results.

Page 9, lines 376-378: “In this sense, the ret-rospective power calculation does not reach 70% (1-β = .67), which should be taken into account when generalising the findings and could be the reason for the non-significance of some of the results.

We thank the reviewer for his comments, as we consider the inclusion of such information essential to make this paper ready for publication.

Round 3

Reviewer 3 Report (New Reviewer)

Thanks very much for providing this information. I think the paper is ready for publication.

This manuscript is a resubmission of an earlier submission. The following is a list of the peer review reports and author responses from that submission.

Round 1

Reviewer 1 Report

The submission is timely and of potential importance to the IDD community. The impact of the covid-related lockdown on the overall mental health of individuals with IDD is an important issue that has not yet been fully investigated or understood.   

As to recommended changes, the submission first requires a comprehensive review with regard to the quality of the English translation. There are sufficient issues with verb tense, word usage, missing or deleted words, etc. that in its current form, the submission is not publishable   In addition:

Line 42 - Please clarify what is meant by "alter respiratory."   

Line 50 - "[O]ur freedom was gone" is a significant overstatement of the issue.

Line 58 - Please clarify what is meant by "in situations of freedom." 

Line 68 - People have an ID; they don't suffer from it. 

Line 70 - Please clarify what is meant by "sanitary attention."

Line 76 - I am unfamiliar with the word "ludic."  Please find a more accessible term. 

Line 112 - If the age range is between 19-37 years, the mean cannot be 40 years. 

Line 133 - Please describe how the San Martino Scale is "totally equivalent to the INICO-FEAPS scale and cite the relevant research. 

Line 135 - There seems to be an omission (cut and paste error?) in the statement "focusing on the type of support required which lets us make [ ] and not on deficits they have."

Lines 149-152 - Please describe the method used to select study participants randomly. 

Line 256 - Why is the paper just targeted at scientific community members? 

General: In most cases, a longitudinal study spans a much longer time than this study does.  Please consider renaming for accuracy. 

General:  More information about why the pre-covid measures were taken is needed.   Are these measures just measures that are typically taken in this setting? Were the authors provided with a warning of a potential lockdown?  

General: Documenting the change resulting from an external independent variable, such as the covid lockdown, without a control group is challenging.  Yes, your pre-lockdown measures act as a control but are a weak control at best. This is made more difficult by the absence of any reliability measures of the 3rd party responders.  Nothing can address this now, but it should be noted in the limitations section. 

My comments were included in the previous section. 

Reviewer 2 Report

-The study contains valuable repeated dataset. However, there are several concerns to be addressed. 

 The longitudinal follow-up typically entails dropout losses which leads to selective sample set. Improved functioning may reflect initially higher baseline functioning participants, which is spurious effect of the current study interest.